# A New Type of Associative Memory Network with Exponential Storage Capacity

## Abstract

Recent developments have sought to overcome the inherent limitations of traditional associative memory models, like Hopfield networks, where storage capacity scales linearly with input dimension. In this paper, we present a new extension of Hopfield networks that grants precise control over inter-neuron interactions while allowing control of the level of connectivity within the network. This versatile framework encompasses a variety of designs, including classical Hopfield networks, models with polynomial activation functions, and simplicial Hopfield networks as particular cases. Remarkably, a specific instance of our construction, resulting in a new self-attention mechanism, is characterized by quasi-exponential storage capacity and a sparse network structure, aligning with biological plausibility. To our knowledge, our proposed construction introduces the first biologically-plausible associative memory model with exponential storage capacity. Furthermore, the resulting model admits a very efficient implementation via vectorization; therefore, it can fully exploit modern numerical computation hardware like GPUs. This work not only advances the theoretical foundations of associative memory but also provides insights into the development of neurobiologically inspired associative memory systems with unprecedented capabilities.

## 1 Introduction

Hopfield networks Amari (1972); Little (1974); Hopfield (1982) are widely recognized as one of the most prominent mathematical models for *associative memory*. In this model, the retrieval of an item is possible by merely recognizing a fragment of its content, such as reconstructing a complete image from a partial view. These models have a rich history which goes back as far as Steinbuch (1961); Willshaw et al. (1969); Longuet-Higgins et al. (1970); Kohonen (1972). Apart from being analytically tractable, Hopfield networks are attractive to biologist because in principle, they can be implemented by the neurons and synapses in the brain. Apart from the necessity of synaptic (neuron-to-neuron) connections, a model can fail to be biologically plausible because of dense connectivity structure whereby, each of the $N$ input neurons is connected to at least a constant fraction $\gamma N$ other nodes. Sparse connectivity means that each input neuron is connected to only a small number of other neurons (i.e, vanishing edge-density in the connectivity graph). This is the case of so-called *dilute* Hopfield networks Derrida & Nadal (1987); Treves & Amit (1988); Bovier & Gayrard (1992); Löwe & Vermet (2011). Finally, it is well-known that matrix models like Hopfield networks have limited *storage capacity* Kohonen (1972); Hopfield (1982); Smolensky (1990); Bovier (1999): they can only store and reliable retrieve $cN$ memory patterns where $c$ is an absolute constant.

The present study introduces a simple yet powerful approach for constructing general Hopfield networks with desirable properties. Our main constributions are summarized as follows.

- *Abstract Hopfield Networks.* Our proposed models utilize setwise connections based on collections, called *skeletons*, of of subsets of input dimensions. We provide analytic expressions for the energy functional and update rule for such models, which extend the definition of traditional Hopfield networks Hopfield (1982). These *Abstract Hopfield Networks* (or AHNs for short), encompass the classical Hopfield network Hopfield (1982), and its various extensions Krotov & Hopfield (2016); Demircigil et al. (2017); Burns & Fukai (2023). Indeed, we will see in Section 2.2 that these models correspond specific choices of

the skeleton. As a by-product, we are able to offer a unified analysis of the storage capacity of the aforementioned networks, and recover known lower-bounds.

- *A New Type Self-Attention Layer.* As our second contribution, we show in Section 3 that a specific choice of skeleton leads to an AHN leads to a new type of self-attention layer which we call *Product-of-Sums Hopfield network (PSHN)* due to its structure. Note that a duality between traditional self-attention layers Vaswani et al. (2017) and a certain type of Hopfield network has also been established Ramsauer et al. (2020). Our proposed PSHN model enjoys the following desirable properties.

  – *Exponential Storage Capacity:* In Section 4 (Theorem 4.1), we show that our PSHN model can robustly store and retrieve up to $e^{c'N}$ memories. We also show (Theorem 4.4) that our network can store and robustly retrieve up to $e^{cN \log^2(N)/\log N}$ patterns. Here, $c$ and $c'$ are absolute positive constants. Thus, the robust storage capacity (formally defined in Section 4) of the model is (quasi-)exponential in the input dimension $N$. Refer to Table 1 for details. Apart from being robust to errors and corruption (e.g due to hardware failure), models with high robust storage capacity integrate long-range interactions in the input domain allowing strongly correlated patterns which characterize the real world, to be correctly separated Ramsauer et al. (2020).

  – *Biological Plausibility.* Finally, in Section 5, we provide strong arguments that our PSHN model can be implemented via two-body sparse synaptic connections between input neurons and hidden a small number of hidden neurons. This makes our model aligned with biology, just like the traditional Hopfield network Hopfield (1982). To the best of our knowledge, our PSHN model is the associative memory model which simultaneously achieves biological plausibility and high storage capacity.

| Type of HN | Reference Paper | Bio Plausible | Robust Storage Capacity |
|---|---|---|---|
| Classical | Hopfield (1982) | Yes | $cN/\log N$ (linear) |
| Polynomial | Krotov & Hopfield (2016) | Yes[†] | $cN^{d-1}/\log N$ (poly) |
| Simplicial | Burns & Fukai (2023) | Yes[*] | $cN^{D-1}/\log N$ (poly) |
| Exponential | Demircigil et al. (2017) | Yes[†] | $\exp(cN)$ (expo) |
| Exponential | Ramsauer et al. (2020) | Yes[†] | $\exp(cN)$ (expo) |
| Little-Hopfield 2 | Hillar et al. (2021) | – | $\exp(\frac{cN}{\log N})$ (quasi-expo) |
| PSHN | Our current work (Section 3) | Yes | $\exp(\frac{cN \log^2 N}{\log N})$ (quasi-expo) |

Table 1: Comparing different types of Hopfield networks (HNs) according to their biological (im)plausibility and robust storage capacity (formally defined in Section 4). For the polynomial (resp. simplicial) Hopfield network, $d$ (resp. $D$) is the degree (resp. dimension). The $c$'s in the exponents of the storage capacity bounds are positive constants which typically depend on the level of robustness required (and also on $d$ and $D$ in the case of polynomial and simplicial Hopfield networks respectively). Yes[†] means the corresponding Hopfield network is only biologically plausible in an indirect sense: it provides an effective description for a more microscopic theory that has additional (hidden) neurons and only requires two-body interactions between them Krotov & Hopfield (2021). Finally, the simplicial Hopfield network Burns & Fukai (2023) only becomes biologically plausible when diluted, i.e a large number of connections are surpressed. This reduces its storage capacity.

**Related Work.** Extensions of Hopfield networks that break the linear of classical Hopfield networks Hopfield (1982) have been proposed. Newman (1988), then Krotov & Hopfield (2016) have shown that a modification of the energy of the classical Hopfield network leads to a polynomial increase in memory capacity. This has been followed up by Demircigil et al. (2017), and more recently Ramsauer et al. (2020) who proposed another modification leading to exponential memory capacity. These so-called modern / dense Hopfield networks use a nonlinear activation function to make the energy and an update rule that is more sharply peaked around the stored memories in the space of neuron's configurations compared to the traditional Hopfield network. See Krotov & Hopfield (2021) for a detailed review. One of the main insights from this recent resurgence of Hopfield networks is their connection to transformers Vaswani et al. (2017), which have become the core components in the design of of large language models (LLMs) for example. Indeed, it was

shown in Ramsauer et al. (2020) that the update / retrieve function in their proposed Hopfield network amounts to the self-attention layer in a transformer model Bahdanau et al. (2015); Cheng et al. (2016); Parikh et al. (2016); Lin et al. (2017); Vaswani et al. (2017). This connection provides hope for a theoretical understanding and explanation of the *emergent* capabilities of modern LLMs.

Hillar et al. (2021) established a duality between certain error-correcting codes on hyper-graphs, and Hopfield networks. This link allowed them to derive an extension of Hopfield networks with quasi-exponential storage capacity. On a similar route, Burns & Fukai (2023) considered an extension of the traditional Hopfield network wherein the complete graph characterizing the connectivity structure of the the neurons is replaced by a simplicial complex. Finally, let us also mention Chaudhuri & Fiete (2019); Smart & Zilman (2021) who have established a direct mapping between Hopfield networks (and extensions thereof) and Restricted Boltzmann Machines (RBMs) Smolensky (1986); Salakhutdinov et al. (2007) whereby the memory patterns of stored by the Hopfield network correspond to parameters that control the activity of the hidden layer in the RBM.

## 2 ABSTRACT HOPFIELD NETWORKS

In this section, we develop a simple and general extension of Hopfield networks. As before, let $N \geq 1$ be the input dimension, that is the number of feature dimensions, or simply input *neurons* (e.g number of pixels in an image). Thus, for simplicity $[N] := \{1, 2, \ldots, N\}$ is the set of (indices of) neurons. As an example, this could be the number of pixels in an image.

A *pattern* is an $N$-dimensional vector $y \in \{\pm 1\}^N$, which encodes a configuration of firing patterns of the $N$ neurons. Let $x^{(1)}, \ldots, x^{(M)} \in \{\pm 1\}^N$ be a set of $M$ distinct patterns we wish to store / memorize. Thus, the $n$th input neuron of the $\mu$th memory is $x_n^{(\mu)} \in \{\pm 1\}$. To specify a Hopfield network, one needs the to specify the energy functional $E : \{\pm 1\}^N \to \mathbb{R}$ and the one-step update run $T : \{\pm 1\}^N \to \{\pm 1\}^N$ for retrieving stored patterns. The $\mu$th pattern is said to be stored or memorized if $T(x^{(\mu)}) = x^{(\mu)}$, i.e if $x^{(\mu)}$ is a fixed point of the update operator $T$. In this study, our focus is exclusively on single-step retrieval, rather than iterative retrieval methods. Opting for one-step retrieval brings several advantages, including the ability to seamlessly integrate the resulting model into a bigger ML model (e.g an LLM) and invoke it during the forward pass.

### 2.1 THE MODEL

Given an arbitrary pattern $y = (y_1, \ldots, y_N) \in \{\pm 1\}^N$ and a subset $\sigma \subseteq [N] := \{1, 2, \ldots, N\}$ of neurons, define a variable $y_\sigma \in \mathbb{R}$ by $y_\sigma := \Pi_{n \in \sigma} y_n$, with the convention that $y_\emptyset = 1$. Note that $y_\sigma$ is a monomial in the variables $y_1, \ldots, y_N$, of total degree $|\sigma|$. For example, if $\sigma = \{1, 5, 7\}$, then $y_\sigma$ is the product $y_1 y_5 y_7$. Analogously to Burns & Fukai (2023), define *coupling constants* $\omega(\sigma)$ by

$$\omega(\sigma) := \sum_{\mu=1}^{M} x_\sigma^{(\mu)}. \tag{1}$$

These quantities will play a similar role as synaptic weights in Hopfield networks Hopfield (1982).

**The Skeleton.** Let $\mathfrak{S}$ be any (nonempty) collection of subsets of $[N]$. We shall call $\mathfrak{S}$ a *skeleton*, borrowing terminology from Burns & Fukai (2023) which considered the special case $\mathfrak{S} = \binom{[N]}{D}$, the collection of all subsets of $[N]$ which contain $D$ or less elements. A skeleton induces a correlation function on $\{\pm 1\}^N$ given by $\langle x, y \rangle_\mathfrak{S} := \sum_{\sigma \in \mathfrak{S}} x_\sigma y_\sigma$ for every pair of patterns $x, y \in \{\pm 1\}^N$. This can also be seen as a feature map as an inner-product in the feature space given by the mapping $y \mapsto (y_\sigma)_{\sigma \in \mathfrak{S}}$. The parameters of the model are the memories $x^{(1)}, \ldots, x^{(\mu)}$, which enter the picture via the coupling constants $(\omega(\sigma))_{\sigma \in \mathfrak{S}}$ given as in (1). For any neuron $n \in [N]$, define

$$\partial_n \mathfrak{S} := \{s \subseteq [N] \mid n \notin s \text{ and } s \cup \{n\} \in \mathfrak{S}\} = \{\sigma \setminus \{n\} \mid \sigma \in \mathfrak{S} \text{ and } n \in \sigma\}. \tag{2}$$

In words, $\partial_n \mathfrak{S}$ is the collection of subsets of $[N]$ which don't contain the neuron $n$ and can be turned into an element of $\mathfrak{S}$ by including $n$. For example, in the case of classical Hopfield networks Hopfield (1982), one has

$$\mathfrak{S} = \{\sigma \subseteq [N] \text{ s.t } |\sigma| = 2\}, \qquad \partial_n \mathfrak{S} = [N] \setminus \{n\} \text{ for all } n \in [N]. \tag{3}$$

Thus, in this case $|\partial_n \mathfrak{S}| = N - 1 \leq N$ for all $n \in [N]$, and we shall see later that this accounts for the linear storage capacity (defined formally in Section 4) of the classical Hopfield network. One can therefore hope to obtain higher storage capacity by appropriate choices for the skeleton $\mathfrak{S}$.

**Energy and One-Step Update Rule.** The energy functional for our abstract model is defined by

$$E(y) := -\sum_{\sigma \in \mathfrak{S}} \omega(\sigma) y_\sigma = -\sum_{\mu=1}^{M} \langle x^{(\mu)}, y \rangle_{\mathfrak{S}}, \text{ for any pattern } y \in \{\pm 1\}^N. \tag{4}$$

The (onse-step) update rule $T : \{\pm 1\}^N \to \{\pm 1\}^N$ is defined component-wise by

$$T_n(y) := \text{sign}\left( \sum_{\mu=1}^{M} x_n^{(\mu)} \sum_{s \in \partial_n \mathfrak{S}} x_s^{(\mu)} y_s \right), \tag{5}$$

for any neuron $n \in [N]$. This construction is a generalization of the energy of the classical Hopfield network Hopfield (1982) by considering arbitrary multi-neuron interactions. For the particular case of classical Hopfield networks Hopfield (1982), it is easy to see from (3) that the energy (4) reduces to $E(y) = -\sum_{\mu=1}^{M} \sum_{n,n' \in [N], \, n' \neq n} x_n^{(\mu)} x_{n'}^{(\mu')} y_n y_{n'}$, while the update rule (5) reduces to update rule $T_n(y) = \text{sign}\left( \sum_{\mu=1}^{M} x_n^{(\mu)} \sum_{n' \in [N], \, n' \neq n} x_{n'}^{(\mu)} y_{n'} \right)$, both of which are well-known formulae.

**Definition 2.1.** *Given a nonempty collection $\mathfrak{S}$ of subsets of neurons $[N]$, the energy (4) and update rule (5) define an Abstract Hopfield Network (AHN) with skeleton $\mathfrak{S}$.*

Thus, once the skeleton $\mathfrak{S}$ is prescribed, everything else about an AHN is completely determined. In particular, when $\mathfrak{S}$ is the collection of subsets of $[N]$ with at most $D$ elements, i.e a simplicial complex of dimension $D$, we obtain the model proposed in Burns & Fukai (2023).

## 2.2 GENERALITY OF OUR CONSTRUCTION

We now show that for specific choices of the skeleton $\mathfrak{S}$, various well-known extensions of Hopfield networks are instances of our AHNs .

**Theorem 2.1.** *The classical Hopfield network Hopfield (1982), the polynomial Hopfield network Krotov & Hopfield (2016); Demircigil et al. (2017), and the simplicial Hopfield network Burns & Fukai (2023),as well as all diluted versions of these networks are all instances of AHNs corresponding to specific choices for the skeleton $\mathfrak{S}$.*

*Proof.* Indeed, as already discussed, for the classical Hopfield network Hopfield (1982), the appropriate skeleton $\mathfrak{S}$ is the collection of all two-element subsets of $[N]$. The polynomial Hopfield networks Krotov & Hopfield (2016); Demircigil et al. (2017) correspond to taking the skeleton $\mathfrak{S}$ to be the collection of all $d$-element subsets of $[N]$. Also, as mentioned earlier, the simplicial Hopfield network Burns & Fukai (2023) corresponds to taking $\mathfrak{S}$ to all subsets of $[N]$ with $\leq D$ elements. $\quad\square$

## 3 THE PRODUCT-OF-SUMS HOPFIELD NETWORK (PSHN)

In this section, we construct an instance of AHNs (Section 2) with remarkable properties like high storage capacity, biological plausibility, and connections to transformers Vaswani et al. (2017).

### 3.1 THE MODEL

**Skeleton of the Model.** Let $G_1, \ldots, G_k$ form a partition of $[N]$. Consider an AHN whose skeleton $\mathfrak{S}$ is the collection of all subsets of $[N]$ which contain exactly one item from each $G_i$, i.e

$$\mathfrak{S} = \mathcal{T}(G_1, \ldots, G_k) := \{\sigma \subseteq [N] \text{ s.t } |\sigma \cap G_i| = 1 \text{ for all } i\}. \tag{6}$$

We call the resulting network a Product-of-Sums Hopfield network (PSHN), a terminology which will become clear later once we make its energy functional explicit. Notice that $\mathcal{T}(G_1, \ldots, G_k)$ can be seen as a bipartite graph with one part (**Left**) consisting of the set of input neurons $[N]$, and the

other part (**Right**) corresponding to the set of group indices $[k]$ acting as hidden neurons; there is an edge between each input neuron $[n]$ and the index $i(n)$ of the group $G_i$ containing $n$. Thus, in this graph, each Left node has degree 1, while the $i$th Right node has degree $N_i = |G_i|$. For fixed groups of equal size $N_i = N/k$ for all $i$, we simply write $\mathcal{T}(N, k)$ for $\mathcal{T}(G_1, \ldots, G_k)$, which then corresponds to a $(1, N_1)$-biregular graph.

The inherent product structure of suc a skeleton enables it to integrate information across long-range interactions among input neurons. For example, we show in Appendix C.3 that these models can solve XOR problem Minsky & Papert (1969), a 3-dimensional problem known to be unsolvable with Hopfield networks comprising fewer than 4 neurons. See also Krotov & Hopfield (2016).

**Energy and Update Rule.**    The energy functional (4) now takes on a special form.

**Lemma 3.1.** *For $\mathfrak{S} = \mathcal{T}(G_1, \ldots, G_k)$, the energy* (4) *is given by $E(y) = \sum_{\mu=1}^{M} E_\mu(y)$, where*

$$E_\mu(y) = -\prod_{i=1}^{k} \sum_{n \in G_i} x_n^{(\mu)} y_n, \text{ for any input pattern } y \in \{\pm 1\}^N. \tag{7}$$

The RHS of (7) justifies the name of the resulting network, namely: Product-of-Sums Hopfied network (PSHN). Figure 5 (Appendix A.3) gives a schematic illustration of (7).

Next, we expand the update rule (5) for the case of our PSHN model. It turns out that the resulting update rule a very admits an efficient algorithmic representation which lends itself well practical implementation as we shall see.

**Lemma 3.2.** *For any pattern $y \in \{\pm 1\}^N$, group index $i \in [k]$, and memory index $\mu \in [M]$, define $a_i^{(\mu)}(y) := \sum_{n' \in G_i} x_{n'}^{(\mu)} y_{n'}$. Then, for any $n \in G_i$, the update* (5) *is $T_n(y) = \text{sign}(\Delta_n(y))$, where*

$$\Delta_n(y) = \sum_{\mu=1}^{M} c_i^{(\mu)}(y) x_n^{(\mu)}, \ c_i^{(\mu)}(y) := \frac{A^{(\mu)}(y)}{a_i^{(\mu)}(y)}, \ A^{(\mu)}(y) := \prod_{j=1}^{k} a_j^{(\mu)}(y). \tag{8}$$

## 3.2    A NEW TYPE OF SELF-ATTENTION LAYER

As already mentioned in the introduction, transformers (aka self-attention layers) are the core component of LLMs. We now show that a specific instance of our proposed PSHN model corresponds to a new type of self-attention layer. Thus our proposed model provides new perspectives for building transformers. So, sonsider the special case where the skeleton is $\mathfrak{S} = \mathcal{T}(N, k)$, i.e where

$$N_i = N_1 = N/k \text{ for all } i. \tag{9}$$

Thus, $N = k \times N_1$. Thanks to Lemma 3.2, the update rule $T$ for our PSHN model admits a simple and efficient implementation based on optimized matrix multiplication (e.g GPUs) that we now describe. Let $X$ be an $M \times N$ matrix whose rows represent $M$ memory patters $x^{(1)}, \ldots, x^{(M)}$, and let $Q$ be an $m \times N$ be a matrix whose rows represent a batch of $m$ input queries $y^{(1)}, \ldots, y^{(m)}$. An implementation of the update rule $T$ is given by the following code in PyTorch Paszke et al. (2017).

```
X = X.reshape((M, k, N1))    # database of memories (e.g clean images)
Q = Q.reshape((m, k, N1))    # incoming queries (e.g noisy/occluded images)
Z = torch.einsum("mkg,Mkg->mMk", Q, X)    # correlate
C = Z.prod(axis=2, keepdims=True) / Z    # this replaces softmax operator
C = torch.nan_to_num(C, nan=0.)    # This is a trick to compute the ci's
TQ = torch.sign(torch.einsum("mMg,Mkg->mkg", C, X))    # output
TQ = TQ.reshape((m, N))    # original shape of input query matrix Q
```

Code Listing 1: PyTorch GPU-friendly implementation of our PSHN model / self-attention layer.

See Appendix A.2 for implementation tips. We come to the realization that in the case of equally sized groups (9), our proposed PSHN model is a new type of self-attention layer schematized in (10), where values $(V)$ = keys $(X)$, and the traditional softmax operator Vaswani et al. (2017) is

replaced with $\sigma$ which maps the $m \times M \times k$ tensor $Z$ to the $m \times M \times k$ tensor $C$ in the the above code snippet. Thus, given query matrix $Q \in \mathbb{R}^{m \times N}$, the model outputs $T(Q) = \text{sign}(\Delta(Q))$, where

$$\underbrace{\Delta(Q)}_{m \times k \times N_1} = \sigma(\underbrace{\widetilde{Q}}_{m \times k \times N_1} \cdot \underbrace{\widetilde{X}}_{M \times k \times N_1}) \cdot \underbrace{\widetilde{X}}_{M \times k \times N_1}, \tag{10}$$

where $\widetilde{X}$ and $\widetilde{Q}$ are reshaped versions of $X$ and $Q$, and '.' is einsum reduction over the second axis.

**A Learnable Version of PSHN Models.** We leave it to future work to experiment extensions of our PSHN models involving embedding matrices, i.e a full-blown self-attention layer of the form

$$X_e = XE_K, Q = QE_Q, V_e = VE_V, \quad \underbrace{\Delta(Q)}_{m \times k \times N_1} = \sigma(\underbrace{\widetilde{Q}_e}_{m \times k \times N_1} \cdot \underbrace{\widetilde{X}_e}_{M \times k \times N_1}) \cdot \underbrace{\widetilde{V}_e}_{M \times k \times N_1}, \tag{11}$$

where $E_K, E_Q, E_V \in \mathbb{R}^{N \times d_e}$, are $d_e$-dimensional embedding matrices for keys, queries, and values respectively, learnable via back-propagation. Here , and $\widetilde{X}_e$, $\widetilde{Q}_e$, and $\widetilde{V}_e$ would be reshaped versions of $X_e$, $Q_e$, and $V_e$, and $(k, N_1, d_e)$ is such that $d_e = k \times N_1$.

## 4 ANALYSIS OF STORAGE CAPACITY

We now turn to the question of storage capacity, i.e how many memory patterns $x^{(1)}, \ldots, x^{(M)}$ can our proposed models store and reliably retrieve ? For this analysis, it is assumed henceforth that the memories are iid, with components $x_n^{(\mu)}$ which are iid Rademacher random variables, so that $\mathbb{P}(x_n^{(\mu)} = \pm 1) \equiv 0.5$, for all $\mu \in [M], n \in [N]$. Let a pattern $y \in \{\pm 1\}^N$ be a pattern obtained from $x^{(1)}$ by forcing a fraction $\theta \in [0, 1)$ of its input neurons to the value $-1$. That is, let $s_\theta \subseteq [N]$ be a uniformly random subset of $\lfloor \theta N \rfloor$ out of $N$ neurons, and for any neuron $n \in [N]$, set

$$y_n^{(\mu)} = -1 \text{ if } n \in s_\theta \text{ and } y_n^{(\mu)} = x_n^{(\mu)} \text{ otherwise.} \tag{12}$$

If $T(y^{(\mu)}) = x^{(\mu)}$, we say that the network has $\theta$-robustly stored $x^{(\mu)}$. When $\theta = 0$ (i.e the nonrobust case), we simply say the network has stored $x^{(\mu)}$. We shall use the following notion of storage capacity which is now standard Hopfield (1982); Bovier (1999); Krotov & Hopfield (2016).

**Definition 4.1** (Robust Storage Capacity). *Given a noise level $\theta \in [0, 1)$, we say a Hopfield network has $\theta$-robust storage capacity $M_{N,\theta}$ if for $M \leq M_{N,\theta}$ and every $\mu \in [M]$, it holds that*

$$\lim_{N \to \infty} \mathbb{P}(\text{The Network } \theta\text{-robustly stores } x^{(\mu)}) = 1. \tag{13}$$

$M_N := M_{\theta,0}$ *is nonrobust storage capacity (i.e for retrieving uncorrupted memories).*

As explained in the introduction (Section 1), the classical Hopfield network Hopfield (1982) only attains linear storage capacity $cN$, for a constant $c$. Over the years, a number of extensions have been proprosed which achieve polynomial Krotov & Hopfield (2016); Demircigil et al. (2017); Burns & Fukai (2023) and even exponential Demircigil et al. (2017); Ramsauer et al. (2020) capacity.

### 4.1 A GENERIC LOWER-BOUNDS FOR ABSTRACT HOPFIELD NETWORKS

We now give capacity bounds for our AHNs (Section 2) in terms of the topology of the skeleton $\mathfrak{S}$.

**Definition 4.2** (Degrees). *The out-degree of a neuron $n \in [N]$ w.r.t to the collection of subsets $\mathfrak{S}$ is defined by $d(n) := |\partial_n \mathfrak{S}|$, where $\partial_n \mathfrak{S}$ is as defined in (2). Also, let $\underline{d}(\mathfrak{S})$ be the minimal out-degree of a neuron w.r.t $\mathfrak{S}$, i.e $\underline{d}(\mathfrak{S}) = \min\{d(n) \mid n \in [N]\}$.*

These are measures of complexity of the skeleton $\mathfrak{S}$ will play a crucial role for storage capacity. The following result (proved in the appendix) is one of our main theoretical findings.

**Theorem 4.1.** *For any AHN with skeleton $\mathfrak{S}$, we have the lower-bound $M_N(\mathfrak{S}) \geq \underline{d}(\mathfrak{S})/(2 \log N)$.*

In the case of nonzero corruption level $\theta \in (0, 1)$, quantitative analysis of storage capacity must exploit further information about the topology of the skeleton $\mathfrak{S}$. Indeed, we cannot generally hope

to get nontrivial lower-bounds for robust storage capacity of an AHN without assumptions on the skeleton $\mathfrak{S}$. For example, the AHN induced by the largest possible collection of subsets of neurons, namely $\mathfrak{S} = 2^{[N]}$, has exponential nonrobust capacity $M_N(\{0,1\}^N) \geq e^{cN}$. This follows from Theorem 4.1 and the fact that $d_{\{0,1\}^N}(n) = 2^{N-1}$ for any $n \in [N]$. However, the basin of attraction around each stored pattern has width zero! To see this, note that $\sum_{\sigma \in 2^{[N]}} x_\sigma y_\sigma = \delta_{x=y}$ by Lemma C.1. Thus, the corresponding energy $E$ in (4) is either 0 or 1. Consequently, it is unable to recover any stored pattern with at a nonzero corruption level, i.e $M_{N,\theta}(\{0,1\}^N) = 0$ for all $\theta \in (0,1)$.

**Definition 4.3** (Moments). *For any $n \in [N]$ and integer $i \geq 0$, define*

$$\mu_{n,i}(\mathfrak{S}) := \max_{s_0 \in \binom{[N]}{i}} |\{s \in \partial_n \mathfrak{S} \mid s_0 \subseteq s\}|. \tag{14}$$

*Thus, there is no subset of $[N]$ with $i$ elements, contained in more than $\mu_{n,i}(\mathfrak{S})$ elements of $\partial_n \mathfrak{S}$.*

Note that in particular, $\mu_{n,i}(\mathfrak{S}) = d(n) := |\partial_n \mathfrak{S}|$. Under the following condition, we can derive a generic lower-bound for the robust storage capacity of an abstract Hopfield network.

**Condition 4.1** (Smooth Skeleton). *(A) $\max_{\sigma \in \mathfrak{S}} |\sigma| - 1 \leq q$ with $q/\log N \to 0$ as $N \to \infty$. (B) $\max_{1 \leq i \leq q} N_1^i \mu_{n,i} = O(d(n))$, for all $n \in [N]$ and some $N_1 \geq N^{c_0}$ and positive constant $c_0$.*

The above condition means for any $i \leq q$, there is no $i$-element subject of $[N]$ which is contained in more than a fraction $N^{-Ci}$ of $s \in \partial_n \mathfrak{S}$, where $C$ is an absolute positive constant.

**Theorem 4.2.** *Fix a corruption level $\theta \in [0,1)$, and consider an AHN with skeleton $\mathfrak{S}$ verifying Condition 4.1. The $\theta$-robust storage capacity is given by $M_{N,\theta}(\mathfrak{S}) \geq c(1-\theta)^{2q} \underline{d}(\mathfrak{S})/\log N$, for some positive constant $c$ which only depends on $\theta$.*

As shown in Section 2.2, the Hopfield network Hopfield (1982); Bovier (1999), polynomial Hopfield networks Krotov & Hopfield (2016); Demircigil et al. (2017), and simplifical Hopfield networks Burns & Fukai (2023) are all instances of our AHNs with appropriate choices of the skeleton $\mathfrak{S}$. Moreover, the skeletons of these models verify Condition 4.1 with $(q, N_1)$ as given in Table 4.1, provided $q = o(\log N)$, i.e $q/\log N \to 0$ as $N \to \infty$.

**Corollary 4.1.** *If $k, d, D = o(\log N)$ as $N \to \infty$, the storage capacity bounds in Table 4.1 hold.*

| Type of HN $\mathfrak{S}$ | $\underline{d}(\mathfrak{S})$ | $q$ | $N_1$ | Robust Storage Capacity |
|---|---|---|---|---|
| Classical | $N$ | 1 | $N$ | $M_{N,\theta} \geq c(1-\theta)^2 N/\log N$ |
| Polynomial | $N^{d-1}$ | $d-1$ | $N$ | $M_{N,\theta} \geq c(1-\theta)^{2(d-1)} N^{d-1}/\log N$ |
| Simplicial | $N^{D-1}$ | $D-1$ | $N$ | $M_{N,\theta} \geq c(1-\theta)^{2(D-1)} N^{D-1}/\log N$ |
| PSHN | $N_1^{k-1}$ | $k-1$ | $N_1$ | $M_{N,\theta} \geq c(1-\theta)^{2(k-1)} N_1^{k-1}/\log N$ |

Table 2: Combined with Theorem 2.1, our Corollary 4.1 recovers lower-bounds previously established in Hopfield (1982); Newman (1988); Krotov & Hopfield (2016); Demircigil et al. (2017); Burns & Fukai (2023). The PSHN model listed in the table is with $k$ equally sized groups (9).

From the corollary, we see that in the small $k$ regime, the storage capacity of our PSHN model with $k$ equally sized groups behaves like that of a polynomial Hopfield network of degree $k$.

### 4.2 STORAGE CAPACITY OF OUR PRODUCT-OF-SUMS HOPFIELD NETWORKS

We now establish lower-bounds for the storage capacity (Definition 4.1) of our proposed PSHN described in Section 3. Let us first consider the case of retrieving a clean / uncorrupted patterns.

**Theorem 4.3.** *The nonrobust storage capacity of the PSHN model with $k$ groups each of size $N_1 = N/k$ verifies $M_N \geq N_1^{k-1}/(2\log N)$. In particular, if $N_1 = O(1)$ in the limit $N \to \infty$, then $M_N \geq e^{cN}$ for some positive constant $c$.*

Figure 1: Theorem 4.3, Empirical Verification ($N = 200$).

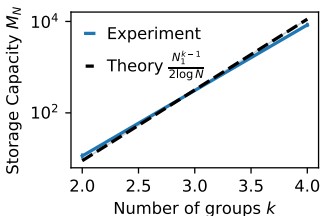

The result is empirically verified in Figure 1. Notice the excellent agreement with experiment. Moreover, it appears the lower-bound is also an upper-bound.

We now turn to robust storage capacity (Definition 4.1). Fix a corruption level $\theta \in [0, 1)$ and let $p := 1 - \theta/2$, $a := 1 - \theta$, and $b := e^{-1/(2a)}$. The next theorem is one of our main results.

**Theorem 4.4.** *Consider the PSHN model with $k$ equal groups each of size $N_1 \geq C \log N$ with $C \geq 73/p$ where $p := 1 - \theta/2$. Then, the $\theta$-robust storage capacity verifies $M_{N,\theta}(\mathfrak{S}) \geq (abN_1)^{k-1}$. In particular, for $N_1 = C \log N$, we have the following lower-bound $M_{N,\theta}(\mathfrak{S}) \geq e^{cN \cdot \frac{\log \log N}{\log N}}$, where $c$ is a positive constant that only depends on $\theta$ and $C$.*

**The Role of the Hyper-Parameters $N_1$ and $k$.** The logarithmic scaling $N_1 \asymp \log N$ is crucial for achieving the quasi-exponential robust storage capacity boasted by our proposed PSHN. For example, if $k = N/N_1$ remains bounded as $N$ goes to $\infty$, then the out-degree $d(n) := |\partial_n \mathfrak{S}| = N_1^{k-1}$ is not large enough to achieve the claimed quasi-exponential lower-bound. On the other hand, if $N_1$ remains bounded in the limit $N \to \infty$, then each group $G_i$ is not large enough to correct for errors and achieve the claimed quasi-exponential lower-bound on storage capacity.

## 5 BIOLOGICAL PLAUSIBILITY OF PSHN MODELS

We now provide strong arguments which show that our proposed PSHN model (Section 3) can be implemented in neurobiology (the brain), at least in principle.

**Synaptic Connections.** Note that our PSHN model can be realized with $O(k)$ hidden neurons for computing sums and products in (8), with direct synaptic connections to input neurons. Unlike the biological plausibility of the "sum" neurons, the "product neurons" which implement $k$-fold multiplication in (8), need some explanation because such an operation might not be implementable biologically by a single neuron. However, this operation can be carried out via a series of $k$ 2-fold / binary multiplication neurons $(a, b) \to a * b$, which are known to be biologically plausible Groschner et al. (2022); Valle-Lisboa et al. (2023). In fact, k-fold multiplication is the basis of so-called sigma-pi networks Feldman & Ballard (1982) and pi-sigma networks Ghosh & Shin (1992), which are well-known in computational neuroscience.

**Sparsity of Connexions.** Concerning the connectivity structure, we see from (8) with $M = 1$ (i.e a single memory pattern) that the graph representing an PSHN model only contains $O(Nk)$ edges in total, corresponding to an edge density of $O(Nk/N^2) = O(k/N)$. If the number of groups $k$ is of order $o(N)$ (e.g $k = O(N/\log N)$), then the computation graph for the corresponding PSHN model is extremely sparse (vanishing edge density), and thus is biologically plausible. Importantly our construction can simultaneously achieve sparsity and (quasi-)exponential robust storage capacity. In contrast, diluted Hopfield networks Derrida & Nadal (1987); Bovier & Gayrard (1992); Burns & Fukai (2023), which enjoy sparsity but only have polynomial storage capacity.

Thus, simultaneously, our proposed PSHN model is biologically plausible and with (quasi-) exponential robust storage capacity. Note that by default, the dense network proposed in Ramsauer et al. (2020) achieves exponential capacity, it is not biologically plausible by design as it requires many-body connections. Krotov & Hopfield (2021) has shown that this network can be seen as an effective description of a more microscropic theory involving only two-body synaptic connections between input and hidden neurons. However, the arguments in favor of the biological plausibility of our model are considerably stronger, as they apply across all scales, not just the microscopic level.

## 6 AN EXPERIMENT: STORING AND RETRIEVING CORRELATED PATTERNS

We empirically demonstrate our theoretical results by running a small experiment on the popular MNIST dataset LeCun et al. (2010). For this computer vision dataset, each of the 70K examples is a gray-scale image of resolution $28 \times 28$ pixels. We sample $M = 10K$ out of 70K images from this dataset and examine how these can be stored and retrieved by our Product-of-Sums Hopfield network (PSHN) model described in Section 3.

**Experimental Setup.** We normalized the intensity values of the image so that they are $\pm 1$. Thus each of the $M = 10K$ images is now a vector in $\{\pm 1\}^N$ with $N = 784$. For each memory pattern

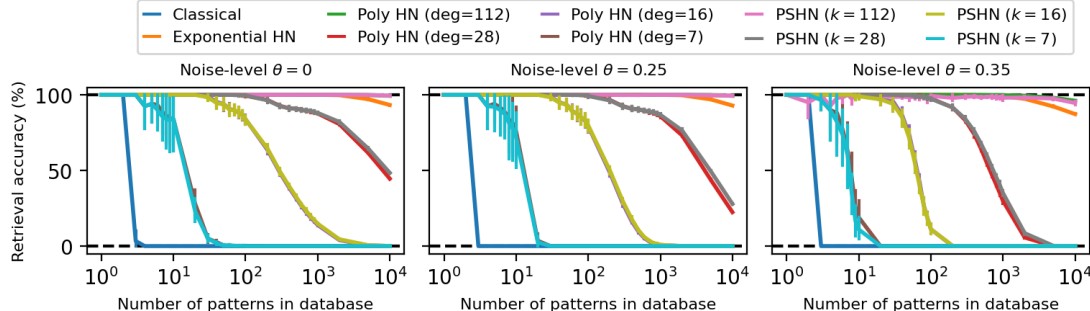

Figure 2: **Comparing storage capacity on MNIST.** Our proposed PSHN model (Section 3) is instantiated with $k$ groups each of size $N_1 = N/k$, where $N = 784$. The $y$-axis represents how many memory patterns are perfectly recovered. Error bars are variations across 10 runs (different sub-samplings of 10K out of 70K images). For this experiment, we see that the optimal number of groups is $k = 112$, each of size $N_1 = N/k = 7$. We also show results for the classical Hopfield network, exponential, and polynomial Hopfield networks discussed in the introduction (Section 1).

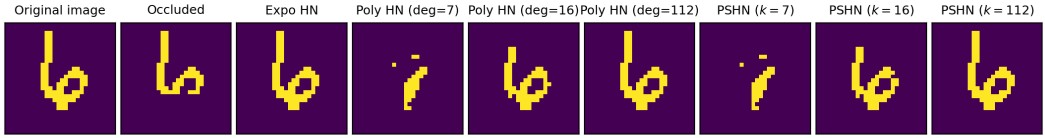

Figure 3: **Visual Inspection** of reconstructed image for each method. As the order of the long-range interactions in the model ($k$ for our PSHN model and degree "deg" for Poly HN) increases, the model moves from feature-extractors to prototype-builders. This is in accordance with the "Feature vs Prototype" theory advocated in Krotov & Hopfield (2016). Appendix A.1 for additional results.

$x^{(\mu)} \in \{\pm 1\}^N$, the pixel intensities of the rightmost $100\theta\%$ portion is set to $-1$. We do this for $\theta = 0$ (corresponding to nonrobust storage), $\theta = 0.25$, and $\theta = 0.35$. We create instances of our PSHN model with $k$ equally sized groups, for different values of $k$ ranging in $\{7, 16, 28, 112\}$. The experiment is run 10 times (on a machine with a single T4 GPU), each time with a different random sub-sampling of $M = 10$K out of the 70K images in the MNIST dataset.

**Empirical Results.** Figure 2 reports robust storage capacity for our model alongside alongside other types of high-capacity Hopfield network discussed in Section 1. Notice how the performance for our PSHN model matches a polynomial Hopfield network of degree $k$, in accordance to Corollary 4.1. For $k = 7$, we observe the best performance for our model, which is consistent with the (quasi-)exponential storage capacity in established in Theorem 4.4. The good performance for the exponential Hopfield network Demircigil et al. (2017); Ramsauer et al. (2020) observed in the figure is also consistent with its exponential storage capacity. These models and ours rely on long-range interactions between features to cope with the strong correlations present in the data. This is unlike the classical Hopfield network Hopfield (1982) which only relies on short-range (pairwise) interactions.

All the models had comparable running times. The entire experiment (10 runs of all models) executes in under 30 minutes on a single T4 GPU. See supplemental for code to reproduce all figures.

## 7 CONCLUDING REMARKS

In this study, we have introduced a versatile framework for extending classical Hopfield networks by incorporating long-range interactions defined via collections of subsets of input features, called "skeletons". We have also demonstrated that many classical Hopfield network extensions are specific examples of our broader construction corresponding to specific choices of the skeleton. Importantly, one specific instantiation of our model introduces a novel self-attention layer (PSHN) with exponential storage capacity. Moreover, we show that the later model is biologically-plausible: it can be implemented by sparse two-body synaptic connections between neurons. Our findings open new possibilities for enhancing machine learning models with powerful associative memory modules.

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

# Appendix

## A    MISCELLANEOUS

### A.1    ADDITIONAL RESULTS: VISUAL INSPECTION FOR RETRIEVAL OF MNIST IMAGES

Figure 4 is a longer version of Figure 2 of the main text (Section 6). See supplemental for code to reproduce all figures.

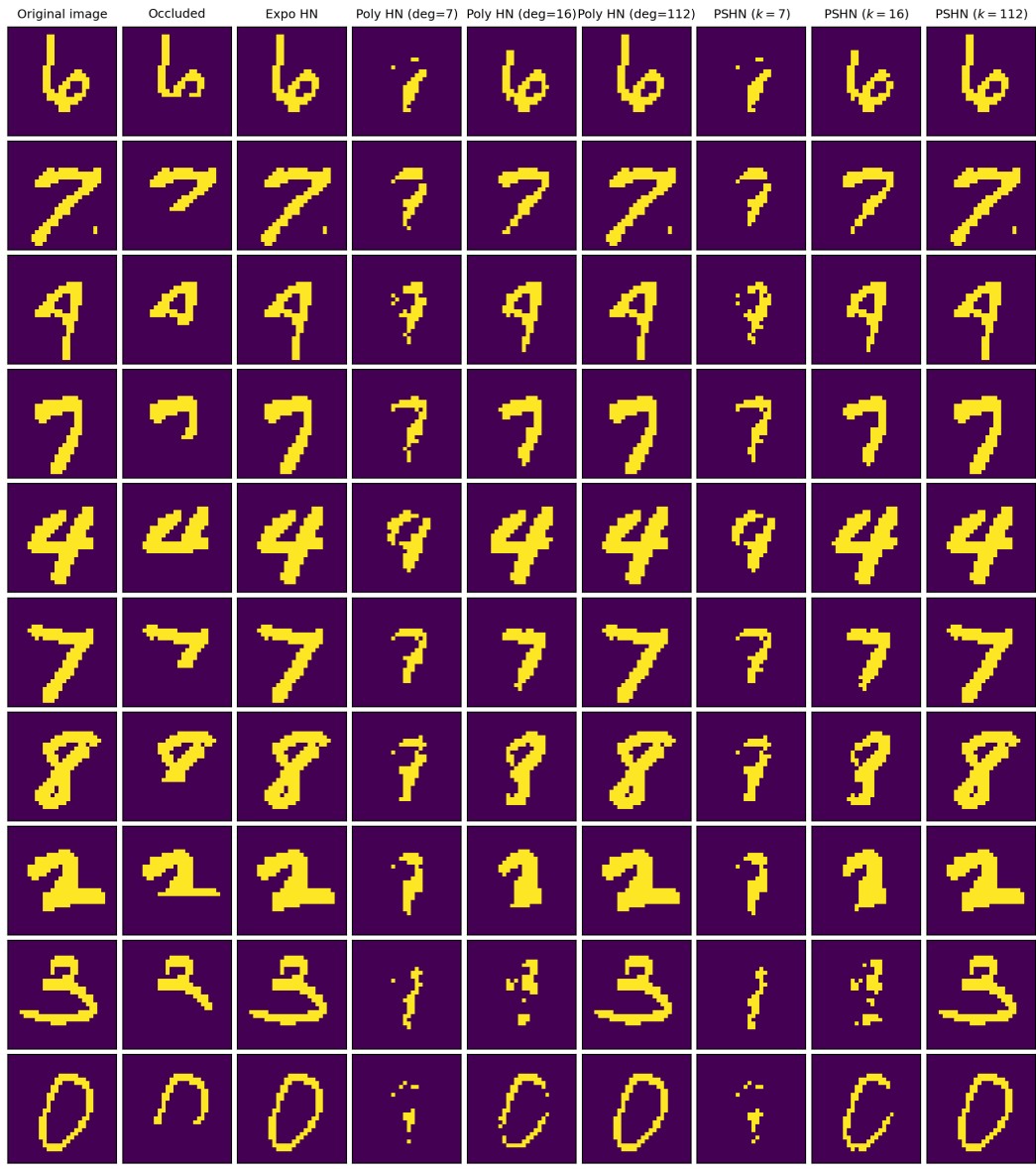

Figure 4: Observe that, as the order of the long-range interactions in the model ($k$ for our PSHN model and degree "deg" for Poly HN) increases, the model moves from feature-extractors to prototype builders. This is in accordance with the "Feature-Extractor vs Prototype" theory advocated in Krotov & Hopfield (2016; 2021).

## A.2 TECHNICAL DETAILS FOR IMPLEMENTING OUR PSHN SELF-ATTENTION LAYER

Observe that the code snippet in Code Listing 1 (Section 3 of the main text) is vectorizable and can take full advantage of optmized linear algebra on GPUs. Also, line 3 of Code Listing 1 is effectively doing matrix multiplication of $k$ pairs of $m \times N_1$ and $M \times N_1$ matrices, and can be carried out in parallel on GPUs, for example. In particular, the case $k = 1$ reduces to the usual matrix product $Z = QX^\top$. A similar comment applies to line 6. All in all, the complexity of our proposed PSHN model is comparable to a traditional self-attention layer Vaswani et al. (2017), and to classical dense associative memory models Krotov & Hopfield (2016); Demircigil et al. (2017); Ramsauer et al. (2020).

## A.3 SOME ILLUSTRATIONS

Figure 5: **Energy Computation for PSHN Model.** Diagram Showing computation of the energy $E(y)$ according to (7) for an input $\pm 1$-pattern $y$ in $N = 9$ dimensions, according to formula (7). Here, there are $M = 3$ memory patterns $x^{(1)}$, $x^{(2)}$, and $x^{(3)}$. Each of the $M$ horizontal blocks of $N$ cells each corresponds to an element-wise product $z^{(\mu)} = x^{(\mu)} \odot y \in \{\pm 1\}^N$, for each $\mu \in [M]$. For this example, the skeleton of the PSHN model is as in (6), with $k = 2$ groups of neurons $G_1 = \{1, 2, 3, 4, 5\}$ and $G_2 = \{6, 7, 8, 9\}$. Each colered subgraph can be seen as a *tokenizer* which correlates the input $y$ and memory patterns $x^{(\mu)}$ along a the input dimensions corresponding to subset of neurons $G_i$. There is an analogous schema (not shown) for computing the update rule $T_n(y)$ defined in (5) according to (8).

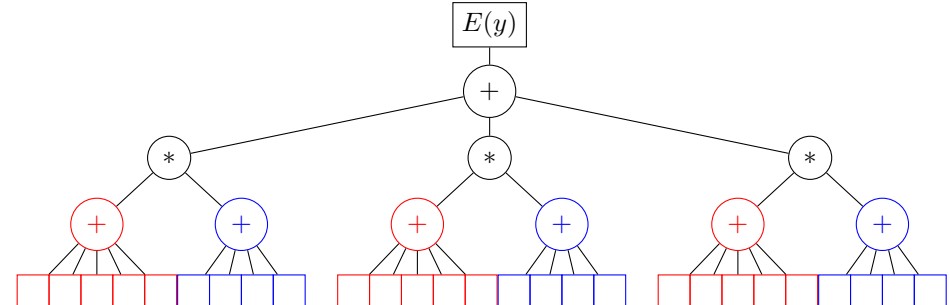

# B PRELIMINARIES ON THE ANALYSIS OF STORAGE CAPACITY

## B.1 NOTATIONS

Let us recall some notations used in the manuscript and define others which will be used in the appendix specifically. We use $[N]$ to denote the set of integers $\{1, 2, \ldots, N\}$. The collection of subsets of $[N]$ with $d$ exactly $d$ elements is denoted $\binom{[N]}{d}$, while $\binom{[N]}{\leq D} := \cup_{d=0}^D \binom{[N]}{d}$ is the collection of subsets of $[N]$ with $D$ elements or fewer. As simplicial complex $K$ on $[N]$ is a collection of subsets of subsets of $K$ such that if $s_0 \subseteq s \in K$, then $s_0 \in K$. For example, $\binom{[N]}{\leq D}$ is a simplicial complex of dimension $D$.

Given nonnegative real functions $f$ and $g$, we write $f(N) \lesssim g(N)$, or equivalently $f(N) = O(g(N))$ to mean that there exists an absolute constant $C$ such that $f(N) \leq Cg(N)$ for sufficiently large $N$, while $f(N) \asymp g(N)$ means $f(N) \lesssim g(N) \lesssim f(N)$. Finally, $f(N) = o(g(N))$, or equivalently $f(N) \ll g(N)$, means $f(N)/g(N) \to 0$ as $N \to \infty$. In particular, $f(N) = O(1)$ means $f$ is bounded, while $f(N) = o(1)$ means $f(N) \to 0$ in the limit $N \to \infty$. For example, $\log N = o(N)$ since $\log(N)/N \to 0$ in the limit $N \to \infty$.

$X \overset{D}{=} Y$ denotes equality in distribution of two random variables $X$ and $Y$.

## B.2 GENERIC SIGNAL TO NOISE RATIO COMPUTATION

Let us consider the problem of robustly storing the first pattern $x^{(1)} \in \{\pm 1\}^N$. Let the pattern $y$ be a corrupted version of $x^{(1)}$ as in (12). That is we will to study the probability that $\mathbb{P}(T(y) = x^{(1)})$. The following argument is adapted from Bovier (1999) which established sharp bounds on the storage capacity of classical Hopfield network (corresponding to $\mathfrak{S} = \binom{[N]}{2}$ in our case). First observe that the update $T_n(y)$ for the the $n$th neuron satisfies $T_n(y)x_n^{(1)} = \text{sign}(\Delta_n)$, where

$$
\begin{aligned}
\Delta_n &= \sum_{\mu=1}^{M} \sum_{s \in \partial_n \mathfrak{S}} x_n^{(\mu)} x_s^{(\mu)} y_s x_n^{(1)} = \sum_{s \in \partial_n \mathfrak{S}} (x_n^{(1)})^2 x_s^{(1)} y_s + \sum_{s \in \partial_n \mathfrak{S}} \sum_{\mu=2}^{M} x_n^{(1)} y_s x_n^{(\mu)} x_s^{(\mu)} \\
&= \sum_{s \in \partial_n \mathfrak{S}} x_s^{(1)} y_s + \sum_{s \in \partial_n \mathfrak{S}} \sum_{\mu=2}^{M} x_n^{(\mu)} x_s^{(\mu)}, \text{ since } x_n^{(1)} y_s x_n^{(\mu)} x_s^{(\mu)} \overset{D}{=} x_n^{(\mu)} x_s^{(\mu)} \text{ (Lemma B.2)} \quad (15) \\
&= \underbrace{A_n}_{signal} + \underbrace{Z_n}_{noise}
\end{aligned}
$$

where the signal term $A_n$ and the noise term $Z_n$ (also known as the *crosstalk* term) are given by

$$
A_n := \sum_{s \in \partial_n \mathfrak{S}} x_s^{(1)} y_s, \tag{16}
$$

$$
Z_n := \sum_{s \in \partial_n \mathfrak{S}} \sum_{\mu=2}^{M} x_n^{(\mu)} x_s^{(\mu)}. \tag{17}
$$

Note that the noise term $Z_n$ as given in (17) is a sum of $(M-1) \cdot |\partial_n \mathfrak{S}| =$ iid Rademacher random variables $x_n^{(\mu)} x_s^{(\mu)}$, and so in terms of (anti)concentration, we expect it to behave like a properly scaled Gaussian random variable. In fact,

**Lemma B.1.** $A_n$ and $Z_n$ are statistically independent, and we have the following identities

$$
\mathbb{E} Z_n = 0, \tag{18}
$$

$$
\text{var}(Z_n) = (M-1)d(n), \tag{19}
$$

$$
\mathbb{E} A_n = \sum_{s \in \partial_n \mathfrak{S}} (1-\theta)^{|s|}. \tag{20}
$$

*In particular, if $\theta = 0$, then $A_n = d(n) := |\partial_n \mathfrak{S}|$, i.e deterministic.*

The hard part of the business is that, in the noisey regime where $\theta = 0$, the variance of $A_n$ will in general depend intricately on the topology of the skeleton $\mathfrak{S}$.

*Proof.* Note that $x_s^{(1)} y_s = \prod_{n' \in s} x_{n'}^{(1)} y_{n'}$, which is a product of $|s|$ idd random variables, each with mean $(1-\theta)$. The claimed expression for $\mathbb{E} A_n$ follows.

We now analyze the noise term $Z_n$. Observe that $Z_n$ and $-Z_n$ have the same distribution. In particular, $Z_n$ has zero mean. Also, the sum in the equation defining $Z_n$ is a sum of $(M-1) \cdot |\partial_n \mathfrak{S}| = (M-1) \cdot d(n)$ iid Rademacher random variables $x_s^{(\mu)} x_n^{(\mu)}$, where and $2 \le m \le M$ and $s \in \partial_n \mathfrak{S}$. We deduce that $Z_n$ has mean 0 and variance given b $\text{var}(Z_n) = (M-1) \cdot d(n)$. $\square$

**Lemma B.2.** *Suppose $n \in [N]$ and $s \subseteq [N]$ such that $n \notin s$. Then, for any memory pattern index $\mu \ne 1$, it holds that $x_n^{(1)} y_s x_n^{(\mu)} x_s^{(\mu)} \overset{D}{=} x_n^{(\mu)} x_s^{(\mu)}$, where $y \in \{\pm 1\}^N$ is obtained from the pattern $x^{(\mu)}$ according to (12).*

## B.3 PROOF OF THEOREM 4.1

Thanks to (15), in order to ensure the "good" event $T_n(x^{(1)}) x_n^{(1)} \ge 0$, we need the std of $A_n + Z_n$, to be dominated by its mean. Thanks to Lemma B.1 we know that in the nonrobust regime ($\theta = 0$),

the former is $\sqrt{(M-1)d(n)}$ and the former is $d(n)$. Thus, it would from Chebychev's inequality that

$$\mathbb{P}(T(x^{(1)}) \neq x^{(1)}) = \mathbb{P}(\exists n \in [N] \text{ s.t } T_n(x^{(1)})x_n^{(1)} \leq 0) \leq \sum_{n=1}^{N} \mathbb{P}(Z_n \geq d(n))$$

$$\leq \sum_{n=1}^{N} \frac{M}{d(n)} \leq N \cdot \frac{M}{\underline{d}(\mathfrak{S})}.$$

This would give the storage capacity bound $M_N(\mathfrak{S}) \gtrsim \underline{d}(\mathfrak{S})/N^{1+o(1)}$, which is sub-optimal with regards to our target. Instead, we use a slightly more involved argument, following a line thought similar to Bovier (1999). First, a standard union-bound gives

$$\mathbb{P}(T(x^{(1)}) \neq x^{(1)}) = \mathbb{P}(\exists n \in [N] \text{ s.t } T_n(x^{(1)})x_n^{(1)} \leq 0)$$

$$\leq \mathbb{P}(\exists n \in [N] \text{ s.t } Z_n \geq d(n)) \leq \sum_{n=1}^{N} \mathbb{P}(Z_n \geq d(n)).$$

Now, for any $n \in [N]$, we know that $Z_n = \sum_{s \in \partial_n \mathfrak{S}} x_n^{(\mu)} x_s^{(\mu)}$ has the same distribution as $\sum_{s \in \partial_n \mathfrak{S}} x_s^{(\mu)}$, because $x_n^{(\mu)}$ and $x_s^{(\mu)}$ are independent. Moreover, the later is a sum of $(M-1) \cdot d(n)$ idd Rademacher random variables $x_s^{(\mu)}$, and so exhibits Gaussian concentration around zero Boucheron et al. (2013). We deduce that

$$\mathbb{P}(T(x^{(1)}) \neq x^{(1)}) \leq \sum_{n=1}^{N} \mathbb{P}\left( \sum_{s \in \partial_n \mathfrak{S}} x_s^{(\mu)} \geq d(n) \right)$$

$$\leq \sum_{n=1}^{N} \exp\left( -\frac{d(n)^2}{2Md(n)} \right) = \sum_{n=1}^{N} \exp\left( -\frac{d(n)}{2M} \right) \tag{21}$$

$$\leq N \cdot \exp\left( -\frac{\underline{d}(\mathfrak{S})}{2M} \right).$$

To make the RHS go to zero in the limit $N \to \infty$, it suffices that $\underline{d}(\mathfrak{S})/M \geq (2+\gamma)\log N$, i.e $M \leq \dfrac{\underline{d}(\mathfrak{S})}{(2+\gamma)\log N}$ where $\gamma$ is a positive constant. We conclude that the storage capacity is lower-bounded as claimed. $\qquad\square$

## C  SOME CALCULATIONS RELATED TO OUR PSHN MODEL

### C.1  PROOF OF LEMMA 3.1

Starting from the general formula (4), we have

$$E(y) = \sum_{\sigma \in \mathfrak{S}} \omega(\sigma) y_\sigma = \sum_{\mu} \sum_{\sigma \in \mathfrak{S}} z_\sigma, \text{ where } z_\sigma := \prod_{n \in \sigma} z_n \text{ and } z_n := x_n^{(\mu)} y_n. \tag{22}$$

Now, by basic algebra, one has

$$\prod_{i=1}^{k} \sum_{n \in G_i} z_n = \sum_{n_1 \in G_1, \ldots, n_k \in G_k} z_{n_1} z_{n_2} \ldots z_{n_2} = \sum_{\sigma \in \mathfrak{S}} z_\sigma, \tag{23}$$

and the result follows upon combing with (22). $\qquad\square$

### C.2  PROOF OF LEMMA 3.2

Indeed, from (5), we know that $T_n(y) = \text{sign}(\Delta_n(y))$, where $\Delta_n(y) := \sum_{\mu=1}^{M} x_n^{(\mu)} \sum_{s \in \partial_n \mathfrak{S}} x_s^{(\mu)} y_s$, where $\partial_n \mathfrak{S}$ is as defined in (2). Now, because $\mathfrak{S} = \mathcal{T}(G_1, \ldots, G_k) := \{\sigma \subseteq [N] \text{ s.t } |\sigma \cap G_j| = 1 \,\forall j\}$, it is clear that if $n \in G_i$, then

$$\partial_n \mathfrak{S} = \{s \subseteq [N] \text{ s.t } |s \cap G_j| = 1 \,\forall j \neq i\} = \mathcal{T}(G_1, \ldots, G_{i-1}, G_{i+1}, \ldots, G_k). \tag{24}$$

We deduce that $\Delta_n(y) = \sum_{\mu} x_n^{(\mu)} \prod_{j \neq i} \sum_{n' \in G_j} y_{n'}^{(\mu)} x_{n'}^{(\mu)} = \sum_{\mu} c_i^{(\mu)} x_n^{(\mu)}$, as claimed. $\qquad\square$

## C.3 Solving the XOR Problem

Let us present the simplest example of a problem which can be solved by our proposed SPS model, but cannot be solved by a classical associative memory model (e.g traditional Hopfield networks Hopfield (1982)) on the same input space: the XOR problem Minsky & Papert (1969). Note that the problem was also considered in Krotov & Hopfield (2016) and shown to be solvable by their polynomial networks as soon as the degree of the polynomial is at least 3. This is because higher-order polynomials induce a capacity limit which surpasses the number of neurons $N$. Indeed, the XOR problem corresponds to $M = 4$ memory patterns of $N = 3$ dimensions given by

$$x^{(1)} = (-1, -1, -1), \ x^{(2)} = (-1, 1, 1), \ x^{(3)} = (1, -1, 1), \ x^{(4)} = (1, 1, -1), \tag{25}$$

with the identification $0 \mapsto -1$ and $1 \mapsto 1$. The 3rd (output) neuron is the XOR of the first two.

Now, consider a partition $\{G_1, G_2\}$ of $[N] = \{1, 2, 3\}$ given by $G_1 = \{1, 3\}$, $G_2 = \{2\}$ and $\mathfrak{S} = \mathcal{T}(G_1, G_2)$. Then, it is easy to see that for any $n \in \{1, 2, 3\}$, then (2) becomes

$$\partial_n \mathfrak{S} = \{\{n'\} \mid n' \in G\}, \ \text{with } G = G_1 \text{ if } n \in G_2; \ G = G_2 \text{ if } n \in G_1.$$

Thus, for any pattern $y \in \{\pm 1\}^3$, the update (5) for the 3rd neuron is $T_3(y) = \text{sign}(\Delta_3(y))$, where

$$\Delta_3(y) = \sum_{\mu=1}^{4} x_3^{(\mu)} \sum_{n \in G_1} z_n^{(\mu)} \sum_{m \in G_2} z_m^{(\mu)} = \sum_{\mu=1}^{4} x_3^{(\mu)} (z_1^{(\mu)} + z_3^{(\mu)}) z_2^{(\mu)}$$

$$= \sum_{\mu=1}^{4} x_1^{(\mu)} x_2^{(\mu)} x_3^{(\mu)} y_1 y_2 + \sum_{\mu=1}^{4} x_2^{(\mu)} y_2 y_3 = -4 y_1 y_2,$$

where we have used the fact that $\sum_{\mu=1}^{4} x_2^{(\mu)} = 0$ and $x_1^{(\mu)} x_2^{(\mu)} x_3^{(\mu)} = -1$ for all $\mu$. Thus, $T_3(y) = -\text{sign}(y_1 y_2) = \text{XOR}(y_1, y_2)$. We deduce that our PSHN model with skeleton $\mathfrak{S} = \mathcal{T}(G_1, G_2)$ solves the XOR problem.

## C.4 A Boolean Binomial Identity

**Lemma C.1.** *For every pair of patterns $x, y \in \{\pm 1\}^N$, it holds that $\sum_{\sigma \subseteq [N]} x_\sigma y_\sigma = \delta_{x=y}$, where $x_\sigma := \prod_{n \in \sigma} x_n$ as usual.*

*Proof.* The proof is by induction on $N$. The case $N = 1$ is trivial since $\sum_{\sigma \subseteq [1]} x_\sigma y_\sigma = 1 + x_1 y_1 = \prod_{n \in [1]} (1 + x_n y_n)$. Suppose the result is true for $N = N'$. We will prove if for $N = N' + 1$. Indeed, observe that

$$\sum_{\sigma \subseteq [N'+1]} x_\sigma y_\sigma = \sum_{\sigma \subseteq [N']} x_\sigma y_\sigma + \sum_{\sigma \subseteq [N']} x_{\sigma \cup \{N'+1\}} y_{\sigma \cup \{N'+1\}}$$

$$= \sum_{\sigma \subseteq [N']} x_\sigma y_\sigma + \sum_{\sigma \subseteq [N']} x_\sigma x_{N'+1} y_\sigma y_{N'+1}$$

$$= (1 + x_{N'+1} y_{N'+1}) \sum_{\sigma \subseteq [N']} x_\sigma y_\sigma$$

$$= (1 + x_{N'+1} y_{N'+1}) \sum_{\sigma \subseteq [N']} \prod_{n \in [N']} (1 + x_n y_n) \ \text{by the induction hypothesis}$$

$$= \prod_{n \in [N'+1]} (1 + x_n y_n),$$

which completes the proof. $\qquad \square$

## D Proof of Theorem 4.2: Storage Capacity of A Class of AHNs

### D.1 Controlling the Signal Term $A_n$ in (15)

We will prove something more general than Theorem 4.2. Let $K$ be a nonempty collection of subsets of $[N]$. Ultimately, we are interested in the case where $K = \partial_s \mathfrak{S}$. Note that $K$ can be seen as an

unweighted hyper-graph with vertex-set $[N]$ and edge-set $K$. Define a random variable $A(K)$ by

$$A(K) := \sum_{s \in K} z_s, \tag{26}$$

where $z_s := \prod_{n \in s} z_n$ as usual. It is clear that the mean of $A(K)$ is given by

$$\mathbb{E}\, A(K) = \sum_{s \in K} (1 - \theta)^{|s|} \tag{27}$$

Let $q = q(K) \geq 1$ be the maximal cardinality of an element of $K$, i.e

$$q(K) := \max_{s \in K} |s|. \tag{28}$$

Thus, the random variable $A(K)$ is a random multi-linear polynomial of degree $q$. Moreover, it is clear that

$$\mathbb{E}\, A(K) \geq (1 - \theta)^q |K|, \tag{29}$$

with equality if $K$ is regular in the sense that $|s| = q$ for all $s \in K$. Now, for any integer $0 \leq i \leq q$, define $\mu_i = \mu_i(K) \geq 0$ by

$$\mu_i := \max_{s_0 \in \binom{[N]}{k}} \sum_{s \in K | s_0 \subseteq s} \prod_{n \in s \setminus s_0} \mathbb{E}\, |z_n| = \max_{s_0 \in \binom{[N]}{k}} |\{s \in K \mid s_0 \subseteq s\}|, \tag{30}$$

where we have used the fact that $|z_n| = 1$, since $z_n$ only takes the values $\pm 1$, for any $n \in [N]$. It is clear that $\mu_0 = |K|$. The other $\mu_i$'s control the size (on average) of the "partial derivatives" of $A(K)$ w.r.t to the elements of $K$. We have the following proposition which is a direct consequence of the main result in Schudy & Sviridenko (2012).

**Proposition D.1.** *With all variables defined as above, it holds for any $\lambda > 0$ that*

$$\mathbb{P}\left(|A(K) - \mathbb{E}\, A(K)| \geq \max_{1 \leq i \leq q} \max(\sqrt{\lambda |K| \mu_i C^q}, \lambda^i \mu_i C^q)\right) \leq e^2 e^{-\lambda}, \tag{31}$$

*where $C \geq 1$ is an absolute constant.*

The appearance of $C^q$ in the result is troublesome and somewhat unavoidable. A very high degree polynomial cannot be concentrated in any meaningful way. Thus, we will focus on the case where the degree $q$ is low in the following sense.

**Condition D.1** (Smoothness). *For some $N_1 \geq 1$ (which may depend on $N, q$) and absolute positive constant $C_1$, it holds that*

$$\max_{1 \leq i \leq q} N_1^i \mu_i \leq C_1 |K|. \tag{32}$$

Note that the above condition only depends on the topology of the underlying collection $K$ of subsets of $[N]$. For example, it is satisfied in the case where $K$ is a simplicial complex on $K_{N, \leq D}$ with $D = O(1)$ (here $(N_1, q) = (N, D)$), or a transversal of an equi-partition partitioning of $[N]$, with $k = O(1)$ groups (here, $(N_1, q) = (N/k, k)$).

**Proposition D.2.** *Under Condition D.1 with $N_1 = N^{\Omega(1)}$ and $q = o(\log N)$ as $N \to \infty$, it holds that*

$$\mathbb{P}\left(\left|\frac{A(K)}{\mathbb{E}\, A(K)} - 1\right| \geq t\right) \leq e^2 e^{-t^2 N^{\Omega(1)}}, \text{ for any } t \in (0, 1). \tag{33}$$

*Proof.* WLOG, take $C_1 = 1$. Observe that

$$\max_{1 \leq i \leq q} \sqrt{\lambda |K| \mu_i C^q} \leq |K| \max_{1 \leq i \leq q} \sqrt{\lambda N_1^{-i} C^q} = |K| \sqrt{C^q \lambda / N_1}. \tag{34}$$

On the other hand, one has

$$\max_{1 \leq i \leq q} \lambda^i \mu_i C^q \leq |K| \max_{1 \leq i \leq q} \lambda^i N_1^{-i} C^q = C^q |K| \cdot \max_{1 \leq i \leq q} (\lambda / N_1)^i$$

$$= |K| C^q \begin{cases} \lambda / N_1, & \text{if } \lambda \leq N_1, \\ (\lambda / N_1)^q, & \text{else.} \end{cases}$$

Thus, for any $t \in (0, C^{q/2})$, taking $\lambda = t^2 N_1/C^q \le N_1$ gives

$$\max_{1 \le i \le q} \max(\sqrt{\lambda |K| \mu_i C^q}, \lambda^i \mu_i C^q) \le |K| \max(\sqrt{C^q \lambda/N_1}, C^q \lambda/N_1) = \max(t, t^2)|K|. \tag{35}$$

Combining this with (31) then gives the following concentration inequality

$$\mathbb{P}\left(|A(K) - \mathbb{E}\,A(K)| \ge \max(t, t^2)|K|\right) \le e^2 e^{-\lambda} = e^2 e^{-t^2 N_1/C^q} = e^2 e^{-t^2 N^{\Omega(1)}}, \tag{36}$$

because $N_1 = N^{\Omega(1)}$ and $q = o(\log N)$ by hypothesis. In particular, taking $t \in (0, 1)$ gives

$$\mathbb{P}\left(|A(K) - \mathbb{E}\,A(K)| \ge t|K|\right) \le e^2 e^{-t^2 N^{\Omega(1)}}, \tag{37}$$

Noting that $\mathbb{E}\,A(K) = \sum_{s \in K}(1-\theta)^{|s|} \ge |K|(1-\theta)^q = |K|N^{o(1)}$ because $q = o(\log N)$, we get

$$\mathbb{P}\left(|A(K) - \mathbb{E}\,A(K)| \ge t\mathbb{E}\,A(K)|\right) \le e^2 e^{-t^2 N^{\Omega(1) - o(1)}} = e^2 e^{-t^2 N^{\Omega(1)}},$$

which completes the proof. $\qquad\qquad\square$

## D.2 PROOF OF THEOREM 4.2

For any neuron $n \in [N]$, applying Proposition D.2 with $K = \partial_n \mathfrak{S}$, $A(K) = A_n$ (the signal term in (15)), and $(N_1, q)$ as in the statement of Theorem 4.2 we obtain that: w.p $1 - O(e^{-t^2 N^{\Omega(1)}})$, it holds that

$$|A_n/\mathbb{E}\,A_n - 1| \le t \text{ with } \mathbb{E}\,A_n = (1-\theta)^q |K| = (1-\theta)^q d(n).$$

Note that the conditions for Proposition D.2 are verified thanks to Lemma D.1. We thus obtain

$$\mathbb{P}(T_n(y) \ne x_n^{(1)}) = \mathbb{P}(T_n(y)x_n^{(1)} \le 0) \le \mathbb{P}(Z_n \ge A_n) \le \mathbb{P}(Z_n \ge (1-\theta)^q d(n)/2) + e^{-N^{\Omega(1)}}. \tag{38}$$

A union-bound in the spirit of the proof of Theorem 4.1 then gives

$$\begin{aligned}
\mathbb{P}(T(y) \ne x^{(1)}) &\le \sum_{n=1}^{N} \mathbb{P}(Z_n \ge (1-\theta)^q d(n)/2) + Ne^{-N^{\Omega(1)}} \\
&= N \cdot \exp\left(-\frac{(1-\theta)^{2q} d(n)^2}{2(M-1)d(n)}\right) + o(1) \\
&= \exp\left(-\frac{(1-\theta)^{2q} d(n)}{2(M-1)} + \log N\right) + o(1),
\end{aligned} \tag{39}$$

and the claimed lower-bound follows. $\qquad\qquad\square$

**Lemma D.1.** *For large $N$ and any positive integer $q \le N$, it holds for any $1 \le i \le q$ that*

$$\mu_i\left(\binom{[N]}{q}\right) = \binom{N-i}{q-i} \lesssim O(N)^{q-i}, \tag{40}$$

$$\mu_i\left(\binom{[N]}{\le q}\right) = \sum_{d \le q} \binom{N-i}{d-i} = O(N)^{q-i}, \tag{41}$$

*where the functionals $\mu_i$ are as defined in (30). Consequently, if $q = o(\log N)$, then $\binom{[N]}{q}$ and $\binom{[N]}{\le q}$ satisfy Condition 4.1.*

## D.3 PROOF OF COROLLARY 4.1

The proof follows from combining Theorem 4.2 with Lemma D.1. We only need to compute $\underline{d}(\mathfrak{S}) := \max_{n \in [N]} |\partial_n \mathfrak{S}|$ for all the networks considered in the corollary.

**Classical Hopfield Networks.** If $\mathfrak{S}$ is the collection all singletons of $[N]$, then $q = 1$ and $\underline{d}(\mathfrak{S}) = N - 1$.

**Polynomial Hopfield Networks.** If $\mathfrak{S}$ is the collection of $d$-element subsets of $[N]$, then $q = d-1$ and $\underline{d}(\mathfrak{S}) = \binom{N-1}{d-1}$. Furthermore, if $N \gg d$, then $\binom{N-1}{d-1} \approx N^{d-1}/d!$.

**Simplicial Hopfield Networks.** The model proposed in Burns & Fukai (2023) corresponds to taking $\mathfrak{S}$ to be a $D$-skeleton on the set of neurons, i.e the collection of subsets of neurons with cardinality $D$ or less, then $q = D - 1$ and $\underline{d}(\mathfrak{S}) = \sum_{d=0}^{D-1} \binom{N-1}{d} \asymp N^{D-1}$. $\qquad\square$

## E    PROOF OF THEOREM 4.3: NONROBUST CAPACITY OF PSHN MODEL

The theorem is a direct consequence of the following lemma.

**Lemma E.1.** *If the subsets $G_1, \ldots, G_k$ with $|G_i| = N_i \geq 1$ for all $i$, form a partitioning of the set of neurons $[N]$, then for the abstract Hopfield network with skeleton $\mathfrak{S} = \mathcal{T}(G_1, \ldots, G_k)$, it holds that $\underline{d}(\mathfrak{S}) = (\prod_i N_i) / \max_i N_i$.*

*Proof.* It is clear that $|\mathfrak{S}| = |G_1 \times \ldots \times G_k| = \prod_{i=1}^{d} N_i$. Now, for any $n \in [N]$ let $G_{i(n)}$ be the unique cluster of neurons which contains $n$. It is clear that $\partial_n \mathfrak{S}$ is isomorphic to $\prod_{i \neq i(n)} G_i$, and so $d(n) := |\partial_n \mathfrak{S}| = \prod_{i \neq i(n)} N_i = |\mathfrak{S}|/N_{i(n)}$, from which it follows that $\underline{d}(\mathfrak{S}) = (\prod_i N_i) / \max_i N_i$ as claimed. $\qquad\square$

*Proof of Theorem 4.3.* For such a partition of $N$, we must have $k = N/N_1 = \Theta(N)$ and so $\prod_i N_i / \max_i N_i \geq 2^k/O(1) \geq e^{\Theta(N)}$ thanks to Lemma E.1. The result then follows directly from Theorem 4.1. $\qquad\square$

## F    PROOF OF THEOREM 4.4: ROBUST STORAGE CAPACITY OF PSHN MODEL

### F.1    WARMUP: A WEAK LOWER-BOUND VIA CHEBYCHEV'S INEQUALITY

Fix $\theta \in [0, 1/2)$. Let $x \in \{\pm 1\}^N$ be uniformly random pattern and let $y \in \{\pm 1\}^N$ be a pattern obtained from $x$ as in (12). Let $d$ and $N_1$ be positive integers and set $N = dN_1$. Partition $[N] := \{1, 2, \ldots, N\}$ $d$ disjoint from $G_1, \ldots, G_d$ of each of size $N_1$, and let $\mathcal{T} = \mathcal{T}(d, N_1)$ be a *transversal* of the $G_i$'s, i.e the collection of subsets of $[N]$ which contain exactly one element from each $G_i$. Note that $\mathcal{T}$ is isormophic to $G_1 \times \ldots \times G_d$ in an obvious way, and thus $|\mathcal{T}| = N_1^d$. Finally, let $z = x \odot y \in \{\pm 1\}^N$ be the component-wise product of $x$ and $y$, and define a random variable $A$ by

$$A(\mathcal{T}) := \sum_{T \in \mathcal{T}} z_T, \tag{42}$$

where $z_T := \prod_{t \in T} z_t$ as usual. Note that $A(\mathcal{T})$ is a *random multilinear polynomial* of total degree $d$. The objective is to design $N_1$ and (thus $d$ too) as a function of $N$ such that $A(\mathcal{T})$ is as large as possible (and positive !) w.p $1 - o(1)$ in the limit $N \to \infty$.

First observe that we can alternately write for every $i \in [d]$,

$$A(\mathcal{T}) = \prod_{1 \leq i \leq d} S_i, \text{ with } S_i := \sum_{t \in G_i} z_t. \tag{43}$$

Now, it is clear that

- The $S_i$'s are iid random variables taking integral values in the range $[-N_1, N_1]$.
- Each $S_i$ is itself a sum of iid random variables which take values $\pm 1$, with $\mathbb{P}(z_t = 1) = 1 - \theta/2$ and $\mathbb{E} z_t = 1 - \theta/2 - \theta/2 = a := 1 - \theta \in [0, 1]$. Thus, $\mathbb{E} S_i = aN_1$, and

$$\mathbb{E} A(\mathcal{T}) = (aN_1)^d. \tag{44}$$

**Proposition F.1.** *In the limit $N_1 \to \infty$ such that $d = o(N_1)$, it holds w.p $1 - o(1)$ that $A(\mathcal{T}) \asymp \mathbb{E} A(\mathcal{T}) = (aN_1)^d$*

*Proof.* Indeed, setting $a := 1 - \theta$, one computes

$$\begin{aligned}
\mathbb{E} S_i^2 &= \sum_{t \in G_i} \sum_{t' \in G_i} \mathbb{E}[z_t z_{t'}] = N_i + \sum_{t, t' \in G_i, \, t' \neq t} \mathbb{E} z_t \mathbb{E} z_{t'} \\
&= N_i + N_i(N_i - 1)a^2 = N_i(1 - a^2) + (N_i a)^2 \\
&= N_i(1 - a^2) + (\mathbb{E} S_i)^2,
\end{aligned} \tag{45}$$

and so $\text{var}(S_i) = N_i(1 - a^2)$. It follows from the independence of the $S_i$'s that

$$\text{var}(A(\mathcal{T})) = \prod_{i=1}^{d} \mathbb{E}\, S_i^2 - \prod_{i=1}^{d} (\mathbb{E}\, S_i)^2 = ((aN_1)^2 + N_1(1 - a^2))^d - ((aN_1)^2)^d$$

$$= ((aN_1)^2)^d \left( \left( 1 + \frac{1/a^2 - 1}{N_1} \right)^d - 1 \right) = (\mathbb{E}\, A(\mathcal{T}))^2 \cdot R(\mathcal{T}), \tag{46}$$

where $R(\mathcal{T}) := \text{var}(A(\mathcal{T}))/(\mathbb{E}\, A(\mathcal{T}))^2 = \left( 1 + \dfrac{c}{N_1} \right)^d - 1$, with $c := 1/a^2 - 1 \geq 0$. Now, one computes

$$0 \leq R(\mathcal{T}) = \left( 1 + \frac{c}{N_1} \right)^d - 1 \leq e^{cd/N_1} - 1.$$

Thus, if $N_1 \to \infty$ such that $d = o(N_1)$ (i.e $d/N_1 \to 0$), then $R(\mathcal{T}) = o(1)$, and Chebychev's inequality gives

$$\mathbb{P}(|A(\mathcal{T}) - \mathbb{E}\, A(\mathcal{T})| \geq (1/2)\mathbb{E}\, A(\mathcal{T})) \leq 4R(\mathcal{T}) = o(1),$$

and the claim is proved. $\qquad\qquad\qquad\qquad\qquad\qquad\qquad\qquad\qquad\qquad\qquad\qquad\qquad\square$

### F.2  A Stronger Lower-Bound Via Chernoff

Let us now remove the troublesome requirement "$d = o(N_1)$" from Proposition F.1. First observe that, in the definition of $S_i$, we can further write $z_t = 2b_t - 1$, where $b_t$ is Bernoulli with parameter $p = p(\theta) := 1 - \theta/2 \in (1/2, 1]$. Thus, $S_i = \sum_{t \in G_i}(2b_t - 1) = 2B_i - N_1$, where $B_i := \sum_{t \in G_i} b_t \sim \text{Bin}(N_1, p)$. By well-known concentration results Boucheron et al. (2013), we have

$$\mathbb{P}(B_i \geq (1+t)N_1 p) \leq e^{-\frac{t^2 p N_1}{2+t}}, \text{ for all } t > 0,$$
$$\mathbb{P}(B_i \leq (1-t)N_1 p) \leq e^{-\frac{t^2 p N_1}{2}}, \text{ for all } 0 < t < 1. \tag{47}$$

We deduce that

$$\mathbb{P}(S_i \geq (2p(1+t) - 1)N_1) \leq e^{-\frac{t^2 p N_1}{2+t}}, \text{ for all } t > 0,$$
$$\mathbb{P}(S_i \leq (2p(1-t) - 1)N_1) \leq e^{-\frac{t^2 p N_1}{2}}, \text{ for all } 0 < t < 1. \tag{48}$$

Therefore: for any $t \in (0, a)$ and $i \in [d]$, it holds w.p $1 - e^{-t^2 p N_1/2}$ that

$$S_i \geq ((2p - 1) - t)N_1 = (a - t)N_1,$$

where $a = a(\theta) := 2p - 1 = 1 - \theta \in (1/2, 1]$ as before. A union-bound over $i \in [d]$ then gives: w.p $1 - \delta(N_1) = 1 - de^{-t^2 p N_1/2}$ it holds that

$$A(\mathcal{T}) \geq (aN_1)^d (1 - t/a)^d = (aN_1)^d ((1 - t/a)^a)^{d/a} \geq (aN_1)^d e^{-td/a} = (ab(t)N_1)^d,$$

where $b(t) := e^{-t/a} \in (0, 1/e)$. Further taking $t = 1/2$ gives: w.p $1 - de^{-pN_1/8}$,

$$A(\mathcal{T}) \geq (abN_1)^d, \tag{49}$$

where $b = e^{-1/(2a)}$. Now, we want $d$ to be as large as possible, and the RHS of the above to be as large as possible too. We can achieve by ensuring that $\delta(N_1) := e^{-pN_1/8 + \log d} \to 0$ in the limit $N_1 \to \infty$. To satisfy this constraint (perhaps non-optimally!) it suffices to take

$$N_1 \geq (9/p) \log d, \tag{50}$$

so that $\delta(N_1) \leq e^{-pN_1/72}$. We have proved the following.

**Proposition F.2.** *If $N_1 \geq (9/p) \log d$, then it holds w.p $1 - e^{-pN_1/72}$ that $A(\mathcal{T}) \geq (abN_1)^d$, where $a := 1 - \theta$, $p := 1 - \theta/2$, and $b := e^{-1/(2a)}$.*

The following result is the last technical step towards the prove of Theorem 4.1.

**Proposition F.3.** *Fix a corruption level $\theta \in [0, 1/2)$ and let $N_1 \geq C \log N$, where $C \geq 73/p$. Then, for large $N$, it holds w.p $1 - o(1/N)$ that*

$$A(\mathcal{T}) \geq (abN_1)^d, \tag{51}$$

*where $a := 1 - \theta$, $p := 1 - \theta/2$, and $b := e^{-1/(2a)}$.*

*Proof.* Indeed, observe that $Ne^{-pN_1/72} = e^{-pN_1/72+\log N} = e^{-(N_1-(72/p)\log N)p/72} = o(1)$ if $N_1 \geq (73/p) \log N$. The result then follows from Proposition F.2 since $\log N \geq \log d$. $\square$

Note that the constant 73 appearing in Proposition F.3 (and also in Theorem 4.4) has not been optimized an could potentially be made much smaller with a bit of more work.

### F.3 PROOF OF THEOREM 4.4

We are now ready to prove Theorem 4.4 proper. Fix a corruption level $\theta \in [0, 1)$, and let $y = y(\theta) \in \{\pm 1\}^N$ be a corrupt version of a memory $x^{(1)}$ which is formed by chosen a subset $s_\theta$ of size $\theta N$ of neurons, uniformly at random and independently of the memories $x^{(1)}, \ldots, x^{(\mu)}$, and then setting

$$y_n = \begin{cases} x_n^{(1)}, & \text{if } n \in s_\theta, \\ -1, & \text{else.} \end{cases} \tag{52}$$

For any neuron $n \in [N]$, the signal term in (15) is given by

$$A_n := \sum_{s \in \partial_n \mathfrak{S}} x_s^{(1)} y_s. \tag{53}$$

Observe that $\partial_n \mathfrak{S}$ is precisely the collection of subsets of $[N]$ which contain exactly one element of each group of neurons $G_i$ except the group which contains the neuron $n$. Applying Proposition F.3 with $\mathcal{T} = \partial_n \mathfrak{S}$, $d = k - 1$, and $A_n = A(\mathcal{T})$ one has $A_n \geq (abN_1)^{k-1} = (ab)^{k-1}d(n)$ w.p $1 - o(1/N)$ as soon as $N_1 \geq C \log N$, where $d(n) = N_1^{k-1}$, $a := 1 - \theta$ and $b := e^{-1/(2a)}$.

Reasoning analogously to (21), we see that

$$\begin{aligned}
\mathbb{P}(T(y) \neq x^{(1)}) &\leq \sum_{n=1}^{N} \mathbb{P}\left(Z_n \geq A_n\right) \\
&\leq \sum_{n=1}^{N} \left(\mathbb{P}\left(Z_n \geq (ab)^{k-1}d(n)\right) + o(1/N)\right) \\
&\leq \sum_{n=1}^{N} \exp\left(\frac{(ab)^{2(k-1)}d(n)^2}{2(M-1)d(n)}\right) + N \cdot o(1/N) \\
&= \sum_{n=1}^{N} \exp\left(\frac{(ab)^{2(k-1)}d(n)}{2(M-1)}\right) + o(1) \\
&\leq N \cdot \exp\left(-\frac{(a^2b^2N_1)^{k-1}}{2(M-1)}\right) + o(1) \\
&= \exp\left(-\frac{(a^2b^2N_1)^{k-1}}{2(M-1)} + \log N\right) + o(1),
\end{aligned} \tag{54}$$

To make the RHS go to zero in the limit $N \to \infty$, it suffices that $(a^2b^2N_1)^{k-1}/M \geq (2+\gamma) \log N$, or equivalently,

$$M \leq \frac{(a^2b^2N_1)^{k-1}}{(2+\gamma) \log N}$$

where $\gamma$ is a positive constant.

In particular, taking $N_1 = C \log N$ and $k = N/N_1$, and then lower-bounding the logarithm of the function $f(N) := (a^2b^2C \log N)^{N/(C \log N)-1}$ by $\Omega(N \log^2 N / \log N)$ gives the result. $\square$

