# OpenReview forum: "A New Type of Associative Memory Network with Exponential Storage Capacity"
_ICLR.cc/2024/Conference — Submitted to ICLR 2024_

### Official Review · Reviewer_CEY5 · 2023-10-23

**Soundness:** 3 good
**Presentation:** 3 good
**Contribution:** 3 good
**Rating:** 5
**Confidence:** 4

**Summary:**

This paper presents the next step in a series of recently-proposed extensions of the classic Hopfield Network, starting with Krotov & Hopfield (2016)'s work on Dense Associative Memories, and followed by the recent work of Burns & Fukai (2023) on Simplicial Hopfield Networks. Their objectives are twofold: to present a general framework that encompasses these past architectures, and to develop within that framework a novel generalization that possesses large storage capacity while remaining biologically plausible.

**Strengths:**

Given the recent interest in associative memory models that generalize the Hopfield network, this paper is timely, and should be of broad interest to the community. In the large, it is reasonably clearly written, and to my knowledge the framework proposed is original. I will caveat my review with the statement that I have for want of time not carefully checked the proofs line-by-line.

**Weaknesses:**

Though I found the paper to be generally interesting, I have a few concerns regarding clarity, and also regarding the claims of biological plausibility. I detail these below under **Questions**.

**Questions:**

1. The critical contribution of Bovier (1999) is the proof of an upper bound on the capacity, showing that the lower bound cited by the authors (which appears in earlier work, for instance Petritis (1996) "Thermodynamic Formalism of Neural Computing") is in fact sharp. The authors prove only lower bounds, which follow from a relatively straightforward Chernoff bound (as in Bovier or in Demircigil's study of polynomial networks). The proof of the upper bound is considerably more technically challenging, requiring a somewhat clever conditional negative dependence argument, and to my knowledge has not been extended beyond the standard Hopfield model. The authors should make this clear when they reference Bovier, and also cite the earlier work of Petritis. It would also be useful to offer some speculation regarding the possibility of proving an upper bound.

2. The empirical test of the predicted capacity in Figure 1 is uncompelling, as the authors test only a single network size. They should probe multiple network sizes, and also provide evidence for the claimed exponential scaling.

3. A key claim of this work is that the proposed model simultaneously enjoys superior biological plausibility compared to previously-proposed associative memory models and has a huge storage capacity. However, I have some concerns regarding their claims. First, they decompose the required $k$-fold multiplication into a sequence of $k$ pairwise multiplications, each of which is claimed to be realized by a neuron. Is their evidence for the existence of this highly structured motif in neural connectivity data? Second, to support the claim that a single neuron can compute a pairwise product, they cite work by Groschner et al. on the fly visual motion detection circuit. Despite the title, what Groschner et al. show is more precisely a form of amplification by disinhibition, which in certain limits closely approximates multiplication (this idea goes back at least to work by Torre and Poggio in 1978, which is the basis for this family of biophysically-inspired motion detection models). How sensitive is their model to departures from perfect multiplication? I can only imagine that the errors would compound over the $k$ approximate multiplications required. This is potentially a severe limitation, as the authors require $k$ to grow with $N$ in order to achieve the promised quasi-exponential capacity.

4. The authors should remind the reader of their notational conventions in Appendix B, particularly the use of set-indexed variables to represent products. In this vein, why do they switch from using $\sigma$ as the index to using $s$?

5. To guide the reader, it would be useful to explicitly compare the argument used here to bound the capacity for polynomial networks to the proof by Demircigil et al. Having a concrete example would in particular make it easier to check each step of the proof quickly.

6. A point of curiosity: Can this framework be easily extended to the hetero-associative memory setting? Several recent works have considered extending Dense Associative Memories to the sequence storage settings, including https://arxiv.org/abs/2306.04532 and https://arxiv.org/abs/2212.05563.

7. It would be useful to mention at least briefly the framework of Millidge et al. (2022), "Universal Hopfield Networks: A General Framework for Single-Shot Associative Memory Models," and contrast it with the framework proposed in the present work.

---

### Official Review · Reviewer_odRF · 2023-10-30

**Soundness:** 3 good
**Presentation:** 3 good
**Contribution:** 3 good
**Rating:** 8
**Confidence:** 4

**Summary:**

The paper proposes an interesting generalization of recently developed dense associative memories and simplicial Hopfield networks with super-linear storage capacity to include product of sums of the input/feature neurons in the definition of the energy. The authors analytically compute the capacity of these models, and study them numerically on correlated patterns.

**Strengths:**

This is a very interesting and natural generalization of the class of models previously discussed by the community. Simple as it is, this has not been done before. Capacity calculation looks plausible, although I have not checked all the details carefully. The illustrations chosen to explain the general construction are also nicely picked and help understanding.

**Weaknesses:**

The paper is generally clear and well-written. The only suggestion that I have is to write down explicitly (in the revised manuscript) the simple model with 9 neurons that is discussed in A3 (both using the energy and the update rule). This would help readers easier understand the general notations used in the main text.

There are also some small typos in the manuscript.

**Questions:**

1. Could the authors please write down explicitly the energy function and update rule for this model with 5+4 neuron model that they discuss in Appendix A3?

2. It seems that the conclusion from Fig 2 is that the behavior of PSHN and Poly HN are roughly similar when deg=k. Is this accurate? If so, might be worth emphasizing this in the text. Also, although it is possible to parse the legend as is, it would be easier for the reader if the items in the legend were grouped in the order they appear in the plot (left to right), e.g. Classical, PSHN=7, PolyHN=7, PSHN=16, etc.

---

### Official Review · Reviewer_f2zF · 2023-10-31

**Soundness:** 4 excellent
**Presentation:** 4 excellent
**Contribution:** 3 good
**Rating:** 8
**Confidence:** 4

**Summary:**

The manuscript presents a new framework for extensions of the Hopfield model for associative memory. A generic "skeleton" defines the allowed interactions in each model, where each previously proposed model corresponds to a specific skeleton (classical model with `1:1` interactions, the polynomial model with rank d interactions, the simplical model with `(D over N)` interactions, etc.). The storage capacity, both robust and nonrobust variants, is analysed for this model family, deriving previously known results. Then, this framework is used to derive a new model (products of sums, or PSHN) where the `N` neurons are divided into `k` disjoint subsets and interactions are allowed only between `k` neurons from different subsets. It is proved that for small `k`, this model has the same capacity as the polynomial model of degree `k`, while the number of interactions is much lower, making the model amendable to efficient implementation (as a new kind of Transformer) and perhaps also biological implementations.

**Strengths:**

* Interesting generalisation of previous results, providing a common ground for comparing them.
 * Proofs of how robust and non-robust storage capacity relate to the structure of the skeleton.
 * Interesting and novel argument showing that for `N/k~log(N)` and small `k`, this model is equivalent to a polynomial model with rank `k` despite a much lower number of interactions (see question).
 * The simulations with up to 10K patterns (with `N=784`) are nice and clean results.
 * Very well-written and a clearly presented argument structure.

**Weaknesses:**

* The biological plausibility argument is under-specified (see questions) and weak. Briefly, a solid argument would explicitly write what are the computations performed by the hidden neurons, how they are connected to the visible ones, and how they are interconnected.
 * No simulation results for the robustness of the memory system with respect for noise (denoted as `theta`). It may be that while the presented method is equivalent to the exponential model with respect to "nonrobust storage capacity", they differ considerably with respect to "robust storage capacity".

**Questions:**

* Can you demonstrate in simulations the similarities and differences between your model and previous ones with respect to "robust storage capacity"?
 * Can you show in a table or in a figure the number of interactions between neurons as a function of `k` for the new model vs polynomial one vs Simplical one? For example, in the conditions used in simulations.
 * For the biological plausibility part, can you explicitly write what would be the computations of the `k` neurons? What would be the connectivity between the `N` neurons and the `k` neurons? Would you have recurrent connections among the `k` neurons? Why did you call it `O(k)` neurons? Those are not at all clear from equation (8)!
 * Figure 2 legend have 10 data items, but only ~5 of them are visible in each panel. Can you use dashed lines or add an indicative data arrow to make it easier to follow?

---

### Official Review · Reviewer_sVrL · 2023-11-02

**Soundness:** 2 fair
**Presentation:** 2 fair
**Contribution:** 2 fair
**Rating:** 5
**Confidence:** 3

**Summary:**

The paper proposes the Abstract Hopfield Network (AHN) (using single step updates and binary memories) that allows the introduction of auxiliary variables ($y_\sigma$) that are the product of aribitrary collections of input neurons. The authors show that this abstraction can describe the classical pairwise Hopfield Network and more complicated structures with multi-neuron interactions.

**Strengths:**

## Abstraction encompasses existing formulations of the Hopfield Net

- (+) Many famous incarnations of the Hopfield Net (e.g., the classical, polynomial, exponential, and simplicial Hopfield Nets) can be expressed in this abstraction
- (+) The model is shown to have a very large storage capacity as the number of feature dimensions increases.
- (+) The paper shows that it is possible to derive something that looks like Transformer's self-attention in this abstraction.



**Originality**: This paper is somewhat novel -- multi-neuron interactions in Hopfield Networks are not new ([Burns & Fukai](https://arxiv.org/abs/2305.05179), [Krotov & Hopfield](https://arxiv.org/abs/2008.06996)), but they present a different framework to understand them and characterize this abstraction.

**Quality and Clarity**: See Weaknesses

**Significance**: See Weaknesses

**Weaknesses:**

## Abstraction is very limited in other ways. Experimental results are minimal and unconvincing

1. (- -) Only studies single-step updates of Hopfield Networks, not recurrent update behavior.
2. (-) Only tested and described as working for binary data.
3. (- -) The paper shows minimal experimental results showing the benefit of this abstraction.
4. (-) Figures to clarify their method are noticeably absent in this writeup
5. (-) There are no experiments to show the effectiveness of the special attention layer.
6. (-) Computational costs/limitations under different choices of skeleton $\mathfrak{S}$ are not discussed
7. (-) Storage capacity tests do not consider the total number of connections, as shown in are communicated in terms of the number of feature neurons, but [Burns & Fukai](https://arxiv.org/abs/2305.05179) showed that it is important to compare storage capacity of different Hopfield Networks with the same total number of connections.


**Originality**: Somewhat novel. See strengths.

**Quality and Clarity**: This paper contains many benign typos and does not strive to make their proposed method accessible by including any kind of descriptive figure.

**Significance**: Medium-Low. The abstraction was not characterized beyond single-step updates and binary memories, and the experimental results and storage capacity do present a noticeable improvement over existing frameworks for understanding Hopfield Networks.

**Questions:**

1. $x_\sigma$ is not defined before Eq (1)?
2. Can you please explain how the AHN explained in this paper can be written in the language of the "separation function" described in [Millidge et al.'s "Universal Hopfield Networks"](https://arxiv.org/abs/2202.04557)?
3. Why are continuous states not studied in this writeup?

Several blatant typos exist in this writeup.:
1. Intro first paragraph: "...attractive to biologist..."
2. pg 1 last paragraph: "...of of subsets..."
3. pg 1 last paragraph: "...correspond specific choices..."

Many more exist, but I stopped recording at this point.

---

### Meta-Review · Area_Chair_q1pk · 2023-12-11

**Metareview:**

The paper proposes an new extension to Hopfield networks with the claim of being the first biologically-plausible associative memory model with exponential storage capacity, and being suitable for modern numerical computation hardware like GPUs.

However, especially reviewers sVrL and CEY5 brought up significant issues. These include insufficient simulations for experimental validation of the paper’s claims, and inadequately supported claims of biological plausibility. The authors did not respond to any of the reviewers' queries.

**Justification For Why Not Higher Score:**

The significant issues raised by the reviewers sVrL and CEY5 were not addressed. See below for these reviews.

**Justification For Why Not Lower Score:**

N/A

---

### Decision · Program_Chairs · 2024-01-16

Reject